# Has the educational use of digital technologies changed after the pandemic? A longitudinal study

**Juan-Ignacio Pozo**[1]☯, **Beatriz Cabellos**[iD][2]☯*, **María del Puy Pérez Echeverría**[1]☯

**1** Department of Basic Psychology, Faculty of Psychology, Autonomous University of Madrid, Madrid, Spain,
**2** Department of Psychology (Evolutionary and Educational Psychology), King Juan Carlos University, Madrid, Spain

☯ These authors contributed equally to this work.
* beatriz.cabellos@urjc.es

**Data Availability Statement:** All relevant data are within the manuscript and its Supporting information files. https://doi.org/10.21950/VYBSEP.

## Abstract

This longitudinal study delves into the impact of the COVID-19 pandemic on the utilization of digital tools in education, particularly focusing on the transition of classrooms into digital spaces. Three years after the lockdown, results from 144 primary and secondary teachers who also participated in a questionnaire during the lockdown indicate an expected decline in the frequency of using digital resources, particularly in attitude and assessment-related activities. Furthermore, activities centred around verbal and procedural learning continue to be teacher-centred. However, notably, experienced teachers using technologies exhibit a relative increase in the adoption of digital technologies for more student-centred activities, underscoring the significance of prior experience in effective technology integration. The study highlights the importance of teacher training to augment the incorporation of digital tools into the curriculum, promoting more effective teaching methodologies.

## 1. The integration of digital technologies in teaching and the effects of the pandemic

For decades, there has been a persistent emphasis on the importance of promoting a genuine integration of digital technologies, also called Information and Communication Technologies (ICT), into the curriculum. It is difficult to believe that the skills required by education in the 21st-century digital societies can be achieved without such integration [1, 2]. The integration goal is to use these technologies as mediators in accessing information and, above all, transforming that information into knowledge [3, 4].

In this regard, the crisis generated by the COVID-19 pandemic marked a turning point in the educational uses of digital technologies. The sudden closure of classrooms due to this health crisis forced teachers worldwide to rely on digital resources as an essential mediator in their students' learning. Therefore, the pandemic represented a critical incident [5–7], with great potential to transform the use of technologies in education. In other words, the pandemic created the conditions for a natural experiment [8, 9], to assess both educational system conditions and any necessary changes.

**Funding:** The contribution of the first author in this article has been supported by a grant from the Ministry of Science and Innovation of Spain (PID2020-114177RB-I00) awarded to JP.

**Competing interests:** The authors have declared that no competing interests exist.

But has there been any change because of this global critical incident in the use of digital technologies? Has their use increased, and more importantly, are they now being used to promote better learning? During the lockdown, we conducted a study on the uses that primary and secondary education teachers were making of digital technologies in their virtual classrooms [7]. In this current research, now that the pandemic is over, we have revisited the same teachers to inquire about these uses, aiming to determine whether there has been a change in their integration into the curriculum.

## 1.1 The use of digital technologies in classrooms before the pandemic

From various perspectives, it has been assumed that ICT can contribute to promoting student autonomy and self-regulated learning [10, 11], facilitating collaborative learning environments, and ultimately fostering the development of 21st-century citizenship skills [1, 2]. However, to achieve this goal, integration needs to be supported by a student-centred strategy [12, 13], whereby with guidance students can o autonomously manage digital resources for learning, promoting constructive rather than reproductive learning. Indeed, studies conducted within this framework in carefully designed and controlled instructional contexts demonstrate that when students manage and make decisions about the use of digital resources, there is a moderate but sustained improvement in learning across different domains [14, 15].

However, these advantages are not consistently maintained across all studies. Numerous international reports, conducted with non-selected samples and in contexts where there is no experimental control over the educational use of digital technologies, such as the PISA reports [16, 17], show more negative results. They even systematically find worse learning outcomes when digital technologies are used in classrooms [18, 19]. Although beyond experimental interventions, there is limited research on how digital technologies are commonly implemented, studies have shown that the most frequent uses are teacher or content-centred, and technological tools are primarily used to present or transmit information [19–21].

From our perspective [7], the differences between data from studies using large samples and experimental works lie in the fact that less specifically selected or previously uninstructed teachers use technologies in a teacher-centred manner. Instead of their students they manage digital resources in the classrooms, especially for presenting or providing content [2, 22]. Thus, before the pandemic, the most common use of digital resources was more teacher-managed than student-managed and was linked to traditional teaching methods, which would explain the limited or even negative effect of these technologies within those massive international studies [7]. But what happened during the pandemic when teachers were massively forced to use digital technologies as the main resource in their classrooms?

## 1.2 The use of digital technologies in classrooms during the pandemic

During the pandemic, numerous studies were conducted on the impact of digitization on education. However, most of them focused on the difficulties of accessing technologies [23], the increase in inequality [5], and the rise of stress, anxiety, and emotional consequences in teaching [24, 25]. In contrast, few studies analysed how digital technologies were being used in the classroom. Among those that did, four types of studies can be distinguished [26].

Firstly, there were prescriptive proposals suggesting how to use ICT [27, 28]. Secondly, questionnaires were applied to understand what activities teachers believed were most suitable [29–31]. Thirdly, entering the classroom, a small number of studies conducted case analyses to observe the activity of selected teachers [32–35]. Similarly to pre-pandemic studies, these case analyses, with very few teachers selected who were highly committed to digital teaching, showed student-centred practices that favoured constructive learning and the acquisition of

digital skills. Studies asking teachers about activities considered most appropriate during that time also tended to identify a student-centred approach.

However, a fourth less common, but more interesting for our objectives type of study investigated the activities teachers were commonly using in their virtual classrooms. The most common form of digital teaching by teachers during the pandemic involved uploading materials to various platforms [22, 30], with most activities being teacher-centred rather than student-centred [7, 26, 30]. Again, there is a gap between what experimental studies and teachers' own beliefs show that can or should be done and what is actually carried out with these technologies. In fact, during the lockdown, in a study on the frequency with which primary and secondary education teachers conducted activities to achieve different learning outcomes (verbal, procedural, and attitudinal), these technologies were mainly used to convey verbal and reproductive knowledge (content or teacher-centred activities), with much less emphasis on procedural or attitudinal content [26]. The only activities directed at attitude learning focused on monitoring work habits in the virtual classroom. Similarly, teachers hardly engaged in activities aimed at managing the emotions produced by the lockdown, despite concerns expressed in other studies about the emotional repercussions of the pandemic on students [36]. The least frequent activities were those supporting cooperative learning. Therefore, the data from that study suggested that during the pandemic, teachers used digital technologies to teach primarily verbal content, with teacher-centred activities, fostering reproductive learning that did not contribute to promoting digital skills. These results were corroborated in an analysis of teaching practices, based on the activities selected and described by the teachers themselves [22, 26].

However, not all teachers used them in the same way. Variables such as gender had little influence, but more complex uses were observed in secondary education than in primary education, especially among teachers who had used these technologies more often before the pandemic. It seems that teachers' use of ICT depends on their training and practice using these technologies, as also noted by Suárez-Rodríguez [20].

Additionally, four teaching profiles related to the use of these tools were identified. Three of them mainly focused on reproductive or teacher-centred learning activities, differing in the quantity of activities they performed. In the fourth profile, there were both teacher-centred and student-centred activities. The use of digital technologies before the pandemic was positively related to this latter profile, again aligning with a more complex usage pattern among teachers with greater experience with these resources.

### 1.3 The use of digital technologies in classrooms after the pandemic

If these were their uses during the pandemic, has this global critical incident changed digital teaching practices? Many authors advocated for such changes [37, 38] and analysed the opportunity it presented for developing more student-centred teaching [39–41]. However, studies that have verified if such changes have occurred are scarce. From an institutional perspective, according to Zancajo et al. [42], the pandemic has contributed to increased and improved access to digitisation, augmenting both technological resources and teachers' knowledge and familiarity with these resources. However, these authors also point out that these changes have not led to modifications in curricula and the orientation of educational policies.

From the teachers' conceptual standpoint, the necessity to work in digital spaces and the acquired experience have contributed to teachers [43] and students [44] showing more positive attitudes towards the use of technologies and asserting that they now use them more than before the pandemic [45, 46], but these studies do not verify if they are actually used. Almerich et al. [47] found that teachers with constructive, more student-centred learning conceptions use digital technologies more in their classrooms than those with more traditional

conceptions. "Losses of learning" have also been measured and analysed in various student cohorts as a consequence of the pandemic years [48, 49].

However, despite all these studies, we have not found any studies that identify if the use of digital technologies has changed since the pandemic. It is the goal of this study through a longitudinal follow-up study, to ask the same teachers from the previous study [7] how they use ICT in their classrooms three years later. Are their practices now more oriented towards constructive learning, or are they still teacher-centred, prioritising content transmission and reproductive learning as we observed during the school lockdown? Has the frequency of activities aimed at achieving verbal, procedural, or attitudinal learning through digital technologies changed? And have been the assessment method and cooperative learning spaces been modified? Which variables are related to the change in teaching practices? Have the teacher profiles changed after the pandemic? These are some questions that have guided our research and are reflected in the following objectives.

### 1.4 Objectives

1. Identify whether there have been changes in the frequency of digital technologies usage among primary and secondary school teachers after the pandemic and whether this frequency is influenced by certain teacher variables (gender, teaching experience, prior usage frequency, educational level, and subject taught).

2. Analyse whether there have been changes in the type of learning promoted (reproductive or teacher-centred vs. constructive or student-centred) by these teachers through digital technologies, as well as the influence of the mentioned variables.

3. Examine whether there have been changes in the learning outcomes targeted by these activities (verbal, procedural, or attitudinal), the type of assessment (reproductive and constructive), and the frequency of cooperative learning, along with the potential influence of the mentioned variables.

4. Investigate whether it is possible to identify teaching profiles in digital technologies usage if these profiles have changed post-pandemic, and how they relate to the other variables under study.

## 2. Method

### 2.1 Task and procedure

To achieve these objectives, we sent the questionnaire electronically, similarly to the previous study [7], to primary and secondary education teachers in Spain who had participated in that study during the lockdown. However, we adapted its content to the current context of face-to-face teaching.

The questionnaire, in addition to requesting personal and professional data and informed consent, comprised 36 items describing different types of activities carried out in the classroom (Table 1). Participants had to indicate the frequency with which they performed these activities on a Likert scale (1, Never; 2, Some days per month; 3, Some days per week; and 4, Every day). These activities differed in whether they were student-centred (constructive) or teacher-centred (reproductive) and whether they promoted verbal, procedural, or attitudinal learning. Activities related to reproductive assessment focused on the product, and constructive assessment, centred on the process, were also included. Finally, cooperative activities were

**Table 1. Dimensions and examples of items.**

| Dimensions | Reproductive activity | | Constructive activity | |
|---|---|---|---|---|
| | N* | Example of items | N | Example of items |
| Verbal learning | 4 | I prepare documents with summaries of the classes and upload them to the platform so that my students can use them when needed. | 4 | I provide my students with various digital materials and links on a topic so they can develop their own perspectives and reflect on them in an assignment. |
| Procedural learning | 4 | I create PowerPoint presentations or other similar resources with instructions on how to complete a task and upload them to a platform so that my students always have access to them and can put them into practice. | 4 | I present an open-ended problem to my students, allowing them to plan their own research or project, search for information on the internet, and carry it out. |
| Attitudinal learning | 4 | I provide my students with links to some pages on study habits to help them organize their work. | 4 | We, as a group, decide on the guidelines for behaviour to better manage virtual communication, and then I upload them to the platform. |
| Assessment | 4 | I create online quizzes for students to complete and then provide them with the correct answers so that, by reviewing their mistakes, they know what needs correction. | 4 | I propose a task to my students and upload all their responses to the platform so that students can appreciate each other's work and justify how it could be improved. |
| Cooperation | | | 4 | I provide my students with various open links and digital materials on a topic for them to work in groups, create a joint report on it, and upload it to the platform. |

N = Number of items per dimension

introduced, which, due to their inherent characteristics, involved a student-centred or constructive approach.

These dimensions were validated through an inter-rater analysis that was conducted during the design of the questionnaire in the previous study [7].

The consistency of the questionnaire and its dimensions was measured through a reliability analysis using the Omega statistic. The reliability of the scale was .955, with the reproductive and constructive scales yielding results above .894 and .935, respectively. The verbal, procedural, attitudinal, evaluation, and cooperation dimensions obtained alphas exceeding .700.

## 2.2 G participants

The questionnaire was sent during May and June of 2023 to the 1403 teachers who had participated in the study three years earlier [7]. On this occasion, we received 193 responses. Questionnaires with duplicate or inconsistent responses with the initial study were removed, leaving a total of 156 responses. From this sample, 12 teachers indicated that they did not use digital resources and did not complete the questionnaire. Therefore, the final sample consisted of 144 teachers. The characteristics of the sample can be seen in Table 2.

To ensure that this sample was a representative subgroup of the 1403 participants from the previous study [7], differences in responses given in the first study by all teachers and by the participants in the current sample were calculated. Absolute differences on all scales were less than 0.10, confirming that the sample in this study did not differ in their initial responses from the rest of the teachers. We obtained the informed consent of all participants and it was approved by the Ethics Committee of the Autonomous University of Madrid.

## 2.3 Analysis of the results

To analyse the first three objectives and compare whether the activities had changed since the lockdown [7] and those obtained in the present study, one-way repeated measures ANOVAs were conducted, which allowed us to identify whether the differences in the amount of activity were significant. Two-way ANOVAs were also performed, one with the same repeated

**Table 2. Characteristics of the sample and variables.**

| Variable | Category | Frequency | % |
|---|---|---|---|
| Gender | Men | 44 | 30.6 |
| | Women | 100 | 69.4 |
| Experience | 10 years of fewer | 46 | 31.9 |
| | From 11 to 20 years | 57 | 39.6 |
| | 21 years or more | 41 | 28.5 |
| Educational level | First years of primary education(6–9 years). | 29 | 20.1 |
| | Last years of primary education(9–12 years) | 29 | 20.1 |
| | Compulsory secondary education (12–16 years) | 62 | 43.1 |
| | Non-compulsory secondary education (16–18 years) | 24 | 16.7 |
| Primary curriculum subjects | Generalists | 28 | 19.4 |
| | Specialists | 26 | 18.1 |
| | Others(missing values) | 4 | 2.8 |
| Secondary curriculum subjects | Social sciences and humanities | 40 | 27.8 |
| | STEM | 31 | 21.5 |
| | Others (missing values) | 15 | 10.4 |
| Previous ICT use | With some frequency | 51 | 35.42 |
| | Habitually | 53 | 36.81 |
| | In all classes | 40 | 27.28 |

measures and another completely randomised, to analyse the influence of variables such as gender, teaching experience, educational level, disciplinary area, and frequency of use of digital technologies. In this way, we were able to observe whether the differences in the amount of activity before and after could be attributed to these variables. The corresponding post hoc analyses were conducted with Bonferroni correction to examine differences between categories.

The Wilcoxon Signed-Rank Test was used to probe into the first objective regarding the self-reported frequency of digital technologies use by teachers. This test considers the order of the data, and was therefore appropriate, taking into account the ordinality of the frequency of use variable. This analysis aimed to verify whether the overall use of technologies had changed from before the pandemic to when the present questionnaire was administered.

Finally, a K-means cluster analysis was implemented to identify different teacher profiles (objective 4). To carry out this analysis, the scores of the different dimensions of the implemented questionnaire were taken into account. Once identified, the Chi-square statistic and its corresponding Corrected Standardized Residuals ($Z_{R(corrected)}$) were used to compare whether there were differences in the number of teachers included in the profiles obtained during the lockdown and the current ones. This allowed us to assess their degree of similarity and assign them a name based on their characteristics. We also used Chi-square and CSR to check whether the new profiles could be related to some of the personal characteristics of teachers (gender, experience, level, area, frequency of ICT use). Lastly, two-way ANOVAs were conducted, one with the same repeated measures used for objectives 1, 2, and 3, and another completely randomised, including the variable generated with teacher profiles. This analysis allowed us to identify whether the differences in the amount of activity before and after the lockdown could be related to the profiles ultimately obtained. Post hoc analyses were performed with Bonferroni correction to examine differences between different profiles and activity changes. All analyses were conducted using SPSS version 28.

## 3. Results

As we have indicated, for our first objective—the longitudinal study of changes in the frequency of digital activity before and after the lockdown—a repeated measures analysis was conducted using data obtained during the lockdown and the results from the present study. Total frequency significantly decreased ($F = 4.35$, $p < .05$, $\eta_p^2 = .03$), as expected when transitioning from exclusively virtual teaching to a context where the use of digital technologies is no longer mandatory. However, a comparison using the Wilcoxon Signed-Rank Test of the frequency with which teachers used digital technologies in a face-to-face setting before and after the pandemic showed a significant increase in usage after the pandemic ($Z = -6.11$, $p < .001$). While 86 teachers increased the frequency of using digital resources, only 23 teachers reduced their usage.

The influence of personal variables was less pronounced after the pandemic. Gender did not yield differences, as in the previous study, and neither did previous teaching experience nor the area of knowledge, in contrast to what was observed during the lockdown. The only variables that now influence are educational level, with increased technology use at higher levels, and, above all, the frequency with which teachers generally use digital resources in their classrooms. Those who use them more also reported higher frequency of different types of activities. The differences are significant ($p < .05$) except for constructive learning of attitudes, which, as we will see, is highly infrequent in all cases. Since they are the only variables with significant effects, they will be considered in the rest of the analyses, disregarding the other variables.

However, the changes are better perceived when we compare the type of use of digital resources. Thus, regarding the second objective of the study—comparing changes based on the type of learning (reproductive or teacher-centred versus constructive or student-centred)– a decrease is observed in reproductive learning ($F = 5.04$, $p < .001$, $\eta_p^2 = .03$) but not in constructive learning ($F = .043$, $p = 837$, $\eta_p^2 2 = 0$) (Fig 1), although this effect is nuanced by the influence of some variables. As we have seen, and will be reiterated in almost all analyses, teachers of the first years of primary education engage in fewer digital activities than the other groups ($p < .001$) and reduce the frequency of more reproductive or teacher-centred activities compared to the lockdown ($p < .001$). Teachers of non-compulsory secondary education decrease in the frequency of teacher-centred activity compared to the lockdown ($p < .05$).

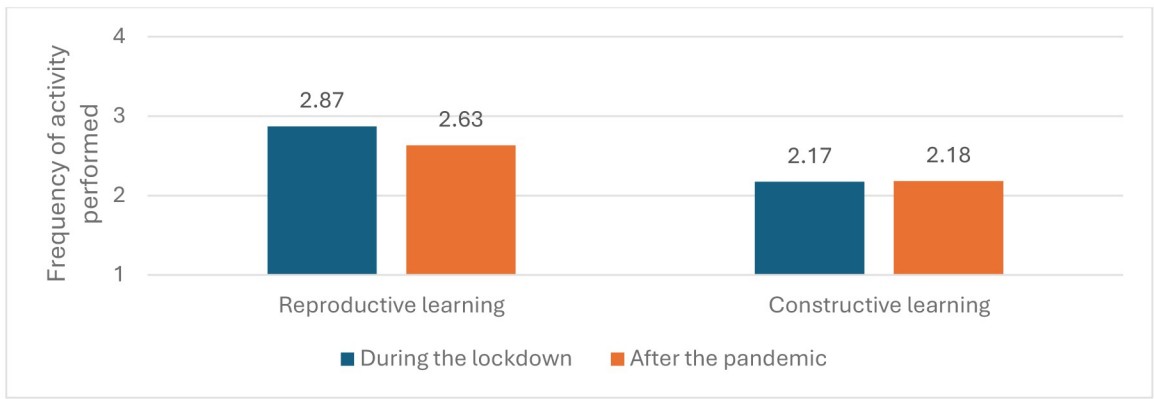

**Fig 1. Differences between reproductive and constructive uses during school lockdown and after the pandemic.** The figure shows that while learning activities significantly decreased after the pandemic, constructive learning activities remained virtually the same in both periods.

The variable that most influences the type of activity performed is the frequency with which teachers use digital resources in general. The two groups with a lower digital activity engage in fewer reproductive or teacher-centred activities than during the lockdown (p < .01), while those using digital resources every day increase student-centred activity (p < .01) and maintain the same teacher-centred activity (p = .154). More interestingly, this group with more frequent use of ICT demonstrate more constructive activity than those who use them less (p < .01).

Analyses related to the third objective, regarding the types of outcomes (Fig 2), showed that the most frequent activities after the pandemic are reproductive verbal and procedural activities, as well as those directed toward reproductive assessment. However, there is a decrease in the frequency of assessment activities, whether reproductive (F = 7.28, p < .01, $\eta_p^2$ = .05) or constructive (F = 13.98, p < .001, $\eta_p^2$ = .09). The frequency of activities aimed at learning attitudes, both constructive (F = 5.94, p < .01, $\eta_p^2$ = .04) and especially reproductive (F = 102.91, p < .001, $\eta_p^2$ = .42), is also lower.

In contrast, the frequency of reproductive verbal activities increase (F = 4.51, p < .05, $\eta_p^2$ = .03), and constructive activities remain the same (F = 1.392, p < .240, $\eta_p^2$ = .10). Additionally, teacher-centred procedural activities decrease (F = 14.63, p < .001, $\eta_p^2$ = .09), but student-centred activities increase (F = 4.53, p < .05, $\eta_p^2$ = .03). Cooperative learning activities also increase significantly (F = 32.72, p < .001, $\eta_p^2$ = .19) (Fig 2).

When analysing the effect of the demographic variables involved, no differences were found between reproductive verbal learning activities during the lockdown and the current moment, except for teachers in compulsory secondary education, where it increases (p < .01). Procedural activities decrease in the first years of primary education and compulsory secondary education, although teacher-centred procedural activities maintain their frequency at all levels, except for the last years of primary education, where an increase is observed (p < .05). As anticipated, there is a general decrease in the frequency of attitudinal activities compared to the lockdown (F = 56.56, p < .001, $\eta_p^2$ = .28), more pronounced in reproductive activities

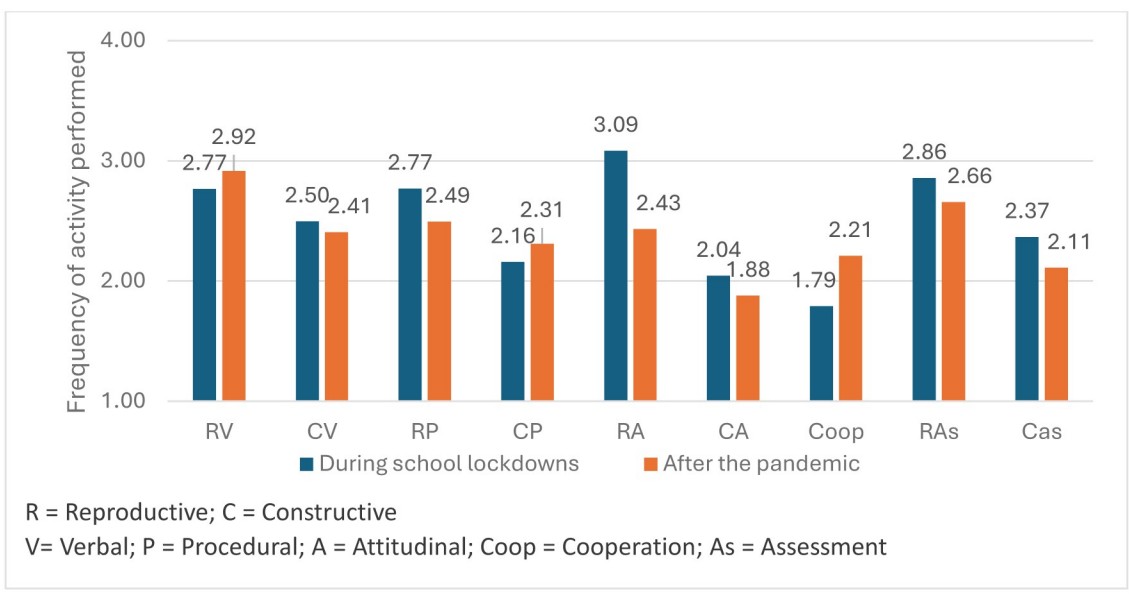

R = Reproductive; C = Constructive

V= Verbal; P = Procedural; A = Attitudinal; Coop = Cooperation; As = Assessment

**Fig 2. Differences between outcomes and types of learning during the lockdown and after the pandemic.** The figure reflects a general trend of decreased frequency in various learning activities after the pandemic compared to during the lockdown, except for reproductive verbal, constructive procedural, and cooperative activities, where there was a significant increase.

(F = 102.91, p < .001, $\eta_p^2$ = .42) than in constructive ones (F = 5.94, p < .01, $\eta_p^2$ = .04), which only significantly reduce in compulsory secondary education (p < .05). These constructive activities are also infrequent during the lockdown, explaining the smaller differences. The frequency of digital assessment activities also decrease after the lockdown (F = 14.917, p < .001, $\eta_p^2$ = .09) in the first years of primary education (p < .001) and non-compulsory secondary education (p < .05), but not in the last years of primary education (p = .841) and compulsory secondary education (p = .138). The same pattern is observed in reproductive assessment. However, constructive assessment decrease in the first years of primary education (p < .001) and compulsory secondary education (p < .05), while the other teachers maintain similar activity. Cooperative learning activities increase significantly after the pandemic (F = 32.722, p < .001, $\eta_p^2$ = .19), especially in the last years of primary education (p < .001) and compulsory secondary education (p < .001), where more cooperative activities are conducted than in the first years of primary education (p < .001). However, once again, the variable that most affects the change in activity is the frequency of use by teachers. Teachers who use them every day engage in more verbal learning activities than before (p < .05). In this group, constructive verbal activities also increase (p < .05), unlike those who make less use, who reduce the frequency of this type of activity (p < .001). The pattern is repeated in procedural activities. Teachers who use them daily engage in more activities than the group with lower use (p < .01). Teacher-centred activities are more frequent in teachers who use digital technologies daily in their classrooms (p < .05) and decrease in the two groups with lower frequency of use (p < .05), as with verbal learning. However, teachers who use the most continue to perform the same number of teacher-centred activities as during the lockdown (p = .154). Activities aimed at attitude learning decrease in all groups (p < .05). This pattern is repeated in the number of reproductive or teacher-centred activities (p < .001), although the decrease is greater in the group that use ICT the least compared to the other two (p < .01). Those who use them every day maintain the frequency of student-centred attitude learning (p = .553). These activities only decrease in the intermediate-use group (p < .001). The use of digital activities for assessment generally decreases in the two groups with less use (p < .05) but not in those who use them in all their classes, and who continued to use these resources during the lockdown (p = .778). This same pattern is observed both in constructive and reproductive assessment. Therefore, in general, while digital assessment activities decrease in groups that use digital resources less frequently in the classroom, they remain unchanged in those who use these technologies more frequently. Cooperative learning activities increase in the group that use them the least (p < .01), but especially in the one that uses them in all their classes (p < .001). However, the only significant differences observed (p < .01), reflect that those who used cooperative activities the least barely engaged in them during the lockdown, and their current activity remains low, despite the increase.

Finally, the fourth objective determined whether teacher profiles in the use of digital technologies could be identified and whether these changed after the lockdown. For this, a cluster analysis was carried out with four teacher profiles obtained from the analysis of current activities performed and this was checked against the 4 teacher profiles identified in the previous study conducted during the lockdown [7]. Afterwards, through chi-square analysis, we determined whether there was a relationship for each teacher between the current profile and the one obtained in the previous study. We observed that one of the three new profiles did not correlate with any of the previous ones ($Z_{R(corrected)}$) < 2.1). This led us to interpret that one of the profiles had been diluted into the other 3: A new cluster analysis of current practices was carried out considering only three profiles. Again, the 4 profiles identified during the lockdown were related through chi-square analysis to the 3 profiles generated with current data. A clear correlation was observed (p < .001) of each of the 3 profiles with the

Passive ($Z_{R(corrected)}$ = 7.1), Active ($Z_{R(corrected)}$ = 5.5), and Interpretative ($Z_{R(corrected)}$ = 10.3) profiles from the previous study [7], but with the Reproductive profile from before not correlating with any of the current profiles. Therefore, we can establish that the current sample groups into three teacher profiles: Passive (composed of 57 teachers), Active (56 teachers), and Interpretative (31 teachers). As seen, by eliminating the Reproductive profile, most of the teachers who composed it are now included in the Passive profile, and to a lesser extent, the Active profile. Only one teacher is added into the Interpretative profile. Once the three profiles had been identified, we analysed whether these they were related to different variables but found no matches, unlike the previous study, where the level of education, area, and previous experience with ICT were related to the profiles [7]. There is, however, a relationship between profiles and the frequency of digital activity (F = 13.84, p < .001, $\eta_p^2$ = .16). Interpretative profile teachers employ ICT more (M = 2.81, SD = 0.57) than the other profiles. In turn, Active teachers (M = 2.42, SD = 0.56) use them more than Passive teachers (M = 2.11, SD = 0.64). Despite these differences, passive teachers are the only ones who maintain the same level of activity as during the pandemic (p = .266), which decrease in the active (p < .05) and interpretative (p < .001) profiles. A more detailed analysis of the type of activities carried out by teachers in each profile helps us understand how their teaching practices have evolved. Reproductive activities decrease in both Interpretative (p < .001) and Active (p < .001) teachers, although both groups engage in more reproductive activity than the Passive profile (p < .001). In the case of constructive learning, student-centred, Interpretative teachers engage in more activity than those in the other two profiles (p < .01), although less than in the previous study, as we have seen. In general, although there are still differences between the profiles in the amount and type of activity, there are fewer differences in the type and amount of activity carried out than in the previous study. The profiles found before and after the lockdown are very similar, although the differences between the profiles have been reduced or blurred a little.

## 4. Discussion and conclusions

During the most intense period of the COVID-19 pandemic, various studies and reports [37, 38] speculated on the psychological and educational consequences of this natural experiment [8, 9]. This was due to the sudden closure of schools and the transformation of all classrooms into virtual spaces mediated by digital technologies. One possible effect would be the acceleration of the integration of ICT into teaching and learning activities. However, there have been few studies that assess, beyond speculation or sparsely representative case studies, the true impact of the pandemic on teaching practices.

In our case, we aimed to evaluate these effects by comparing the activities that teachers carried out during the COVID-19 lockdown, using digital resources, with those same teachers almost 3 years later. This type of study, as far as we know, is new in this field. While there are numerous longitudinal follow-up studies on the emotional and psychosocial effects of COVID-19 [50, 51], there is no similar work analysing how teaching and learning practices have changed as a result of the widespread and mandatory use of digital technologies in confined classrooms.

As expected, the results have shown that, after returning to normalcy with face-to-face classes where digital resources may or may not be used, teachers use these technologies less frequently than during the lockdown when they were the only way to maintain their teaching activity. However, the results show that, nevertheless, they use them more than before the pandemic. We can say, therefore, that the mandatory use of digital technologies during the pandemic has facilitated their educational inclusion. Despite the unfavourable conditions under

which their mandatory use was imposed, there is greater acceptance of digital technologies among teachers than before the pandemic. This result is similar to findings in other studies [39, 46], which also show that access to technologies has been facilitated post-pandemic [42, 52].

However, the study has not been limited to our first objective of comparing changes in the quantity of digital practice. Building on the theoretical framework outlined in the Introduction and on the design of the types of activities proposed in the original study [7], we have been able to verify which digital teaching activities have increased or decreased in frequency after the pandemic, addressing our second objective. We have also compared whether these activities promote more reproductive learning, from a teacher-centred approach, or foster constructive learning, where students have control over digital resources. According to our data, as it happened during pandemic lockdown, it is mainly teachers who manage these resources in the classroom, promoting reproductive learning. As shown in Fig 2, the most frequent activities are still those oriented towards reproductive learning, whether verbal or procedural, as well as the reproductive assessment of these learnings.

This trend was observed not only in the previous study [7] but also in other studies during COVID-19 [30, 33] and even in other studies before the pandemic [17, 20, 21]. However, this traditional dominance of teacher-centred use is more moderate in the present study than in the previous one, as several results indicate a more selective use of digital resources after the pandemic.

In this sense, within the mentioned general pattern, there is a greater decrease in teacher-centred activities, while activities oriented towards constructive learning maintain their frequency. This implies a higher relative frequency of student-centred activities after the pandemic, especially noticeable in the case of constructive procedural activities. The increase in activities based on student cooperation ($\eta_p^2$ = .19) is particularly significant, considering it was the least implemented type of activity during the lockdown [7]. Cooperative activities are considered one of the educational uses of ICT that can most improve learning, due to their potential to foster dialogical spaces [2, 3, 10]. After the pandemic, these activities have increased the most in frequency, although they remain in the minority.

This trend toward a more selective use of digital technologies after the pandemic is also observed when analysing our third objective, related to the types of learning promoted (verbal, procedural, or attitudinal) and their use for assessment. In fact, digital resources are currently used the least for attitudinal learning and assessment. While digital activities aimed at verbal and procedural learning are generally maintained—the results of learning for which these resources are most used (Fig 2)—there is a radical decrease in the use of ICT to promote student-centred attitudinal learning, especially behaviour control through teacher-centred uses. Assessment with digital resources, whether reproductive or constructive, also decreases, albeit to a lesser extent. It can be assumed that attitudinal and behavioural learning, as well as assessment, have not disappeared from classrooms but are managed mainly in face-to-face spaces rather than through digital activities. The focus of digital use has shifted more toward conveying verbal and procedural knowledge (reproductive verbal and procedural learning), and to a lesser extent, fostering understanding and strategic knowledge management (constructive verbal and procedural learning).

The case of assessment is particularly relevant, as the reluctance to conduct digital assessments reflects difficulties in their integration into the curriculum. Teachers can use ICT to design learning activities, but changing their assessment methods by introducing digital technologies is more challenging. In this format, traditional assessments measuring students' accumulated knowledge beyond the use of certain formats like Kahoot or Google Forms [53] are not possible. Integrating digital technologies into assessment largely requires a shift toward

constructive assessment goals and designing tasks that go beyond content reproduction. During the pandemic, one of the teachers' concerns was to prevent cheating, which is crucial in reproductive assessment. Resources were even created to address this issue [54, 55]. It is not coincidental that now, with the impact of generative Artificial Intelligence in education, one of the main concerns of teachers is how it affects assessment systems [56], as it questions traditional assessment methods based on the accumulation of knowledge.

However, the decrease in the use of digital technologies, in addition to concentrating on certain activities more than others, also focuses on certain educational levels. Indeed, digital activity decreased significantly in the first years of primary education (6–9 years) and partly in non-compulsory secondary education (16–18 years), but not in the rest of the educational levels, where its use has been maintained post-pandemic. In the case of younger children, during the lockdown, virtual activities were usually carried out at home with the support of family, due to the children's lack of proficiency. It is understandable that, with no family mediation, teachers resort less to these activities in the classroom.

Regarding the studied teacher variables, neither teaching experience nor disciplinary area had any effect on the results, unlike the previous study [7]. Gender did not produce any differences either then or now. Of these variables, in addition to the educational level, the only variable that affects how teachers use digital technologies in the classroom is precisely the degree to which they are accustomed to doing so. Those who use digital technologies daily in their classrooms tend to make more complex, student-centred uses now than during the lockdown, both in verbal and procedural activities and especially in cooperative ones. These teachers even maintain the same use as during the lockdown in activities that, as we have seen, show a marked decrease in the rest of the teachers, such as teaching attitudes and assessment.

Therefore, not all teachers use these technologies in their classrooms equally, nor have they changed their usage patterns equally after the pandemic. The practice or expertise in the educational use of ICT seems to be a critical variable, not only in the frequency of use but especially in how or for what purpose it is used [57]. However, the last objective of our study was to analyse whether distinct patterns or profiles can indeed be identified among teachers and how they have changed after the pandemic.

Although the teacher profiles have been reduced from four to three in the present study (Passive, Active, and Interpretative), it is interesting to note that they are still maintained over time. In general, teachers are associated with the same profile as in the previous study, although the teachers who were part of the Reproductive profile then are now distributed between the Active and Passive profiles, both equally oriented toward reproductive learning. The new profiles correlate with the profiles obtained in the previous study, reflecting stability in the conceptions or positions maintained by teachers. However, in the present study, differences between the different profiles in the type of activities carried out through digital technologies have diminished. We would say that although different teacher profiles still exist, there are fewer differences in the activities they perform, although new research would be necessary to deepen in the meaning of these differences.

Thus, although the three teacher profiles essentially maintain the type and amount of activity compared to the lockdown, Passive profile teachers tend to engage in more digital activity now than before. The other two profiles remain more digitally active than the Passive profile, although their level of activity has decreased compared to the lockdown. However, it should be noted that the Interpretative profile is still the one that performs the most student-centred activities, showing significant differences from the other two groups.

Against this more limited influence of the profiles, prior experience with digital resources seems to better explain the differences found in this study, as identified by Peng, Razak, and Halili [58]. The results show that this experience not only makes it more likely to use them in

classrooms but also increases student-centred uses, which, in turn, promotes better learning. Therefore, to increase their integration into the curriculum, a greater and better use in teacher training should be encouraged, both for practising teachers and those in training [59]. The more teachers become accustomed to their mediating role in learning, the more likely they are to make student-centred uses, which promote greater curricular integration of these technologies and generate better learning [2].

In conclusion, although uses are still more reproductive than constructive, hindering the educational integration of digital technologies, the longitudinal follow-up after the pandemic shows a closer approach to these integration goals, especially among those teachers who use them daily in their classes.

## 4.1 Limitations of the study

This study innovatively contributes to the analysis of the consequences of the COVID-19 pandemic for education by longitudinally tracking a group of teachers three years after the school lockdown. However, one common limitation of such longitudinal studies is the loss of sample between one wave and another [60]. Although we have been able to confirm that the reduced sample in this second wave is representative of the larger sample obtained in the first wave, we would have undoubtedly preferred a larger participant recruitment. Unfortunately, various external variables beyond our control (such as changes in educational institutions or discontinuation of teaching activities) significantly reduced the sample size, which may have led to the reduced influence of some of the studied variables on the results.

A second limitation of this study relates to questionnaire use. Despite efforts to closely align with teachers' actual practices rather than just their general beliefs about how those practices should be [26], the survey still relies on what teachers say they do rather than necessarily capture what they do in their classrooms. We are aware of the typical gap between how teachers conceptualize or perceive their activity and their actual practices [61–63]. Therefore, it would be interesting to complement these studies with analyses of real teaching practices [22, 26], as well as studies identifying the relationship between teachers' conceptions and their practices in the use of digital technologies [47, 64, 65].

Lastly, it is relevant to indicate the limitations of the K-means analysis [66]. Although it is a popular technique for identifying clusters, this analysis has limitations such as dependency on the initialization of centroids, bias caused by outlier cases, and the need to choose a fixed number of clusters. Although this work attempted to mitigate these potential issues by performing clustering with different data orderings, starting with the means of the dimensions rather than the raw items, and checking the differences between the selection of different numbers of clusters, it is important to keep these limitations in mind and consider this test as merely an exploratory analysis of the possible teacher profiles identified. In any case, it is important to highlight the correlations between the obtained profiles and those from the previous study, which may indicate some consistency between the results obtained through this analysis.

## Supporting information

**S1 Checklist. Human participants research checklist.**
(DOCX)

## Acknowledgments

We would like to appreciate Krystyna Sleziak for her support in preparing the English version of the paper.

## Author Contributions

**Conceptualization:** Juan-Ignacio Pozo, María del Puy Pérez Echeverría.

**Data curation:** Beatriz Cabellos.

**Formal analysis:** Beatriz Cabellos.

**Funding acquisition:** Juan-Ignacio Pozo.

**Investigation:** Juan-Ignacio Pozo, Beatriz Cabellos, María del Puy Pérez Echeverría.

**Methodology:** Juan-Ignacio Pozo, Beatriz Cabellos, María del Puy Pérez Echeverría.

**Project administration:** Juan-Ignacio Pozo.

**Resources:** Juan-Ignacio Pozo, Beatriz Cabellos.

**Software:** Beatriz Cabellos.

**Validation:** Beatriz Cabellos.

**Visualization:** Beatriz Cabellos.

**Writing – original draft:** Juan-Ignacio Pozo, Beatriz Cabellos, María del Puy Pérez Echeverría.

**Writing – review & editing:** Juan-Ignacio Pozo, Beatriz Cabellos, María del Puy Pérez Echeverría.

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
