## [Decision Letter · Decision Letter 0]

22 Aug 2024

PONE-D-24-15719Has the educational use of digital technologies changed after the pandemic? A longitudinal StudyPLOS ONE

Dear Dr. Cabellos,

Thank you for submitting your manuscript to PLOS ONE. After careful consideration, we feel that it has merit but does not fully meet PLOS ONE’s publication criteria as it currently stands. Therefore, we invite you to submit a revised version of the manuscript that addresses the points raised during the review process. Reviewers are generally positive about your article and asked for some changes. Beside these,I urge you to take the following into consideration.Elaborate more on the statistical tests conducted, the data suitability and evaluation of themAlso, it is important to discuss the limitation of these tests, k-means and what it means for your findings.More importantly, be sure to align your findings and conclusions with your results without offering any conclusions not supported by your data

We look forward to receiving your revised manuscript.

Kind regards,

Mohammed Saqr, Ph.D

Academic Editor

PLOS ONE

“The contribution of the fist author in this paper was supported by a grant from the Spanish Ministry of Science & Innovation (PID2020-114177RB-I00).”

“The contribution of the fist author in this paper was supported by a grant from the Spanish Ministry of Science & Innovation (PID2020-114177RB-I00). We would like to appreciate Krystyna Sleziak for her support in preparing the English version of the paper.”

“The contribution of the fist author in this paper was supported by a grant from the Spanish Ministry of Science & Innovation (PID2020-114177RB-I00).”

4. In this instance it seems there may be acceptable restrictions in place that prevent the public sharing of your minimal data. However, in line with our goal of ensuring long-term data availability to all interested researchers, PLOS’ Data Policy states that authors cannot be the sole named individuals responsible for ensuring data access (http://journals.plos.org/plosone/s/data-availability#loc-acceptable-data-sharing-methods).

Reviewers' comments:

Reviewer's Responses to Questions

**Comments to the Author**

1. Is the manuscript technically sound, and do the data support the conclusions?

Reviewer #1: Yes

Reviewer #2: Yes

2. Has the statistical analysis been performed appropriately and rigorously? 

Reviewer #1: Yes

Reviewer #2: Yes

3. Have the authors made all data underlying the findings in their manuscript fully available?

Reviewer #1: Yes

Reviewer #2: No

4. Is the manuscript presented in an intelligible fashion and written in standard English?

Reviewer #1: Yes

Reviewer #2: Yes

5. Review Comments to the Author

Reviewer #1: The paper presented is original in its content and provides relevant data for the field of educational technology.

The theoretical framework of justification of the proposal as well as the methodology used is relevant and the results are significant.

Can be published

Reviewer #2: I congratulate the authors for the clarity of the text and the precision in the presentation of the objectives, the methodological design and the results. On the other hand, the conclusions are coherent with the results obtained and the discussion is correct, although it could be enriched with some references linked to this topic that have been previously published in PlosOne on teaching practices with ICT just before the confinement and closure of schools. The topic is of great interest for Educational Technology research and the approach taken is very timely to increase knowledge about the evolution of digital education after the pandemic.

Although the PLOS Data policy requires authors to make available all data supporting the findings described in their manuscript without restriction and the Data Availability Statement in the PDF file of the manuscript indicates that ‘Yes - all data are fully available without restriction’, we have not found how to access these data. Access to the full questionnaire is also missing. Knowledge of the instrument is very important for a better understanding of the results and is a good practice in relation to open knowledge.

With regard to the questionnaire used in the research, information on its reliability is provided, but no information on the validity of the instrument is included.

The two main limitations of the study are correctly identified. On the one hand, the considerable loss of sample in the replication of the questionnaire after its use during the pandemic. On the other hand, the need for triangulation of data through qualitative methodologies that allow us to understand the phenomenon in greater depth. There are many studies based on the teacher's subjective view of his or her own activity that should be completed with other data collected in the context and with other views of significant agents (students, families).

In conclusion, this is a relevant study, well designed and developed, with interesting results and well-constructed conclusions based on the evidence obtained. It is recommended that the discussion be improved with some references to teaching practices with ICT pre-COVID and information on the validity of the instrument. Finally, consistent with the Data Availability Statement, provide access to the survey instrument and data.

6. PLOS authors have the option to publish the peer review history of their article (what does this mean?). If published, this will include your full peer review and any attached files.

Reviewer #1: No

Reviewer #2: No

---

## [Author Response · Author response to Decision Letter 0]

6 Sep 2024

Dear Editor,

We would like to express our sincere gratitude to you and the reviewers for your valuable feedback on our manuscript titled “Has the Educational Use of Digital Technologies Changed After the Pandemic? A Longitudinal Study.” Below, we have outlined the suggestions provided by both you and the reviewers:

Editor’s Suggestions:

1. Method:

o It was suggested to delve deeper into the statistical tests performed, the appropriateness of the data, and its evaluation.

Thank you very much for your suggestion. In section 2.3, "Analysis of the results," we have tried to delve deeper into the statistical tests performed as indicated.

2. Discussion:

o The editor advised discussing the limitations of the K-means analysis and what these limitations imply for the findings. 

Thank you very much for the recommendation. In section 4.1, "Limitations of the study," we have addressed the limitations of the K-means analysis.

o It was also recommended to ensure that the findings and conclusions are aligned with the results, without offering any conclusions not supported by the data. 

We greatly appreciate your suggestion. We have included a few adjustments on page 22 of the manuscript to improve readability.

3. Other Recommendations:

o We were asked to clarify the role of the funders in the study, and if they had no role, to explicitly state that in the cover letter. 

As the editor recommended, we have added the following statement: "The funders had no role in study design, data collection and analysis, decision to publish, or preparation of the manuscript."

o We were informed that funding information should appear in the Cover Letter and not in other sections of the manuscript.

We have removed the funding information from the manuscript and included it in the cover letter.

o The editor suggested providing a non-author contact for data access requests to ensure long-term availability of the data.

We have added an institutional link where both the questionnaire used and the dataset can be found, which will make it easier for readers to replicate the data: https://doi.org/10.21950/VYBSEP

o The inclusion of separate legends for each figure in the manuscript was requested.

We have included separate legends for each figure as recommended.

o We were also advised to review the references, and any changes made should be noted in the letter to the editors and reviewers.

We have added the following references to the manuscript, but no changes have been made to the previous references:

• Valverde-Berrocoso J, Fernández-Sánchez MR, Revuelta Dominguez FI, Sosa-Díaz MJ. The educational integration of digital technologies pre-Covid-19: Lessons for teacher education. PLoS One. 2021;16(8). https://doi.org/10.1371/journal.pone.0256283

• Peng R, Abdul Razak R, Halili SH. Factors influencing in-service teachers’ technology integration model: Innovative strategies for educational technology. PLoS One. 2023;18(8). https://doi.org/10.1371/journal.pone.0286112

• Vasuthevan K, Vaithilingam S, Ng JWJ. Academics’ continuance intention to use learning technologies during COVID-19 and beyond. PLoS One. 2024;19(1). https://doi.org/10.1371/journal.pone.0295746

• Ahmed, M, Seraj, R, Islam, SMS. The k-means algorithm: A comprehensive survey and performance evaluation. Electronics.2020; 9(8): 1-12. https://doi.org/10.3390/electronics9081295

Reviewer 1’s Suggestions:

Reviewer 1 expressed full satisfaction with the manuscript.

We sincerely thank Reviewer 1 for their positive feedback and full satisfaction with our work. We are pleased that the manuscript meets the high standards expected, and we appreciate their recognition of the effort and rigor that went into this research.

Reviewer 2’s Suggestions:

1. Conclusions:

o The reviewer suggested that while the conclusions are consistent with the results, the discussion could be enriched by including references related to similar studies, particularly those published in PLOS ONE concerning teaching practices with ICT before the lockdown.

As indicated to the editor, we have added a series of references published in PLOS ONE that help justify the discussion of the results obtained.

2. Ethics:

o The reviewer noted that although the Data Availability Statement indicated that all data were fully accessible, there was no clear method to access these data, including the complete questionnaire. The reviewer emphasized the importance of making the full instrument available for better understanding of the results.

We have added an institutional link where both the questionnaire used and the dataset can be found, which will make it easier for readers to replicate the data: https://doi.org/10.21950/VYBSEP

3. Method:

o The reviewer pointed out that while information about the reliability of the questionnaire was provided, information regarding the validity of the instrument was missing and should be included.

We greatly appreciate the reviewer's suggestion. In section 2.1, "Task and procedure," we have added information regarding the validity of the instrument.

We appreciate the detailed feedback provided, which will help us to refine and improve our manuscript.

Yours faithfully,

The authors of the manuscript.

---

## [Editor Report · Decision Letter 1]

24 Sep 2024

Has the educational use of digital technologies changed after the pandemic? A longitudinal Study

PONE-D-24-15719R1

Dear Dr. Cabellos,

We’re pleased to inform you that your manuscript has been judged scientifically suitable for publication and will be formally accepted for publication once it meets all outstanding technical requirements.

Kind regards,

Mohammed Saqr, Ph.D

Academic Editor

PLOS ONE
---

## [Editor Report · Acceptance letter]

2 Oct 2024

PONE-D-24-15719R1 

PLOS ONE

Dear Dr. Cabellos, 

I'm pleased to inform you that your manuscript has been deemed suitable for publication in PLOS ONE. Congratulations! Your manuscript is now being handed over to our production team.

Kind regards, 

on behalf of

Dr. Mohammed Saqr 

Academic Editor

PLOS ONE